# Antioxidant, anti-prostate cancer potential, and phytochemical composition of the ethyl acetate stem bark extract of *Boascia coriacea* (Pax.)

**Gervason Moriasi**[1,2]*, **Mathew Ngugi**[1], **Peter Mwitari**[3], **George Omwenga**[1]

**1** Department of Biochemistry, Microbiology and Biotechnology, School of Pure and Applied Sciences, Kenyatta University, Nairobi, Kenya, **2** Department of Medical Biochemistry, School of Medicine, Mount Kenya University, Thika, Kenya, **3** Centre for Traditional Medicine and Drug Research, Kenya Medical Research Institute, Nairobi, Kenya

* gmoriasi1@outlook.com

**Data Availability Statement:** All relevant data are within the manuscript and its Supporting information files.

## Abstract

### Background

The antioxidant and anticancer potential of natural compounds, particularly from medicinal plants, is increasingly being explored as alternatives to synthetic antioxidants and chemo-therapeutics. *Boascia coriacea* (Pax) has been traditionally used for treating various ailments, including oxidative stress-related diseases and prostate cancer. However, there is a paucity of empirical evidence to validate the ethnomedicinal claims, hence this study.

### Methods

The antioxidant capacity of the extract was assessed using 1,1-Diphenyl-2-picryl Hydrazyl radical (DPPH) radical scavenging and hydrogen peroxide scavenging assays, alongside total antioxidant capacity. *In vitro* cytotoxicity was determined using the 3-(4,5-dimethylthia-zol-2-yl)-2,5-diphenyltetrazolium bromide (MTT) assay on Vero CCL-81 normal cells and DU-145 prostate cancer cells. Gene expression levels of *ar*, *bcl-2*, *caspase 3*, *cdk1*, and *p53* were quantified using qPCR to elucidate the mechanisms of action. Phytochemical analysis was conducted using gas chromatography-mass spectrometry.

### Results

The studied plant extract exhibited significant DPPH radical scavenging activity, with an $EC_{50}$ of 0.008 µg/ml, 10-fold lower than that of L-ascorbic acid (0.08 µg/ml), indicating potent antioxidant capacity. Similarly, the extract demonstrated substantial hydrogen peroxide scavenging activity, albeit with lower efficacy ($EC_{50}$ of 1039.10 µg/ml) compared to L-ascorbic acid, and a total antioxidant capacity of 454.39±25.26 µg AAE/mg dw. *In vitro* cytotoxicity assay revealed a $CC_{50}$ of 68.61 µg/ml against Vero CCL-81 cells and an $IC_{50}$ of 32.16 µg/ml against DU-145 cells, with a superior selectivity index of 2.13, compared to doxorubicin's 1.46. The extract significantly downregulated the expression of *ar*, *bcl-2*, normalised

**Funding:** This study was partly supported by the German Academic Exchange (DAAD) scholarship for PhD Study granted to Gervason Moriasi under the In-Country/In-Region Scholarship Programme (Ref: 91843013). There was no additional external funding received for this study.

**Competing interests:** The authors have declared that no competing interests exist.

*caspase 3*, *cdk1* genes while upregulating *p53* in DU-145 cells, suggesting its role in inducing apoptosis and inhibiting cancer cell proliferation. Phytochemical analysis identified 19 compounds, including lup-20(29)-en-3-one (7.99%) and lupeol (59.49%), which are associated with anticancer activity.

## Conclusion

The ethyl acetate stem bark extract of *B. coriacea* demonstrates significant antioxidant and anticancer activities, potentially through modulation of apoptosis and cell cycle pathways. The presence of bioactive compounds supports its potential as a therapeutic agent, warranting further investigation for developing novel treatments for prostate cancer and oxidative stress-related conditions.

## 1. Introduction

Oxidative stress (OS), is a fundamental factor in the pathogenesis of numerous debilitating conditions, including PCa [1]. The body's intrinsic enzymatic and nonenzymatic antioxidant defence mechanisms typically counterbalance these pro-oxidants to maintain redox homeostasis. However, during disease states or excessive exposure to pro-oxidants, these defences are overwhelmed, leading to sustained oxidative stress and the progression of various diseases, including cancer [2]. Synthetic antioxidants such as butylated hydroxyanisole (BHA), tertiary butylhydroquinone (TBHQ), butylated hydroxytoluene (BHT), and propyl gallate (PG) are currently employed to bolster endogenous defences against OS [3]. Despite their effectiveness in scavenging free radicals, their extensive use has been linked to adverse health effects, including carcinogenicity, cytotoxicity, endocrine disruptions, and immunological disorders. Given the significant role of oxidative stress in prostate cancer (PCa) and the limitations of synthetic antioxidants, there is a critical need for alternative antioxidants from natural sources that are both safe and effective in mitigating OS and its associated pathologies.

Cancer remains one of the leading causes of mortality worldwide, ranking as either the first or second leading cause of death in most countries. Patients often succumb to the disease prematurely, before the age of 70 years [4]. Prostate cancer, in particular, present significant health challenges, with high morbidity and mortality rates affecting across the globe [5]. Sub-Saharan African countries, including Kenya, bear a disproportionate cancer burden, with over 80% of cancer cases attributed to limited access to quality healthcare In Kenya, cancer is the second leading cause of disease-related deaths, following cardiovascular disorders [6]. Besides, research indicates a higher mortality rate among PCa patients of African descent compared to their white counterparts, influenced by genetic, socioeconomic, and environmental factors [7]. This disparity is largely due to ineffective healthcare systems, the high cost of cancer treatment, and limited resources for accessing quality care [8]. Additionally, conventional anticancer therapies often lead to severe side effects, therapeutic failure, and high recurrence rates [9], underscoring the urgent need for alternative, safer, and more affordable treatments.

*Boscia coriacea* (Pax), a woody shrub native to Kenya and known locally by various names, is an important plant in traditional medicine and ethnoveterinary practices. Its stem bark, roots, and leaves are used to treat ailments such as body pains, ear problems, and respiratory infections, as well as veterinary conditions like anthrax [10]. Recent phytochemical studies have identified several bioactive compounds in *B. coriacea*, including alkaloids, flavonoids,

glycosides, and tannins, which exhibit antimicrobial, anti-inflammatory, antioxidant, and anti-cancer activities [10]. However, comprehensive scientific studies on its antioxidant and anti-cancer properties, safety, and detailed phytochemical profile are limited. Therefore, this study aimed to investigate the cytotoxicity, anti-prostate cancer, antioxidant activity, and phyto-chemical composition of the ethyl acetate stem bark extract of *B. coriacea*, thereby validating its traditional uses and exploring its potential as a source of new therapeutic agents against oxidative stress and associated diseases, such as prostate cancer.

## 2. Materials and methods

### 2.1 Plant material

Fresh stem barks of mature *B. coriacea* were collected from their natural habitat in Mbeere North Sub-County, Embu County, Kenya as described by previously [11]. The selection of this plant for this study was based on its ethnomedicinal use by the local community for managing disorders associated with oxidative stress, especially cancer of the prostate. Initially, this plant was identified by a local herbalist by its local name as *"Mutangira"*. The plants were taxonomically authenticated by a qualified botanist (NMK/BOT/CTX1/2), and duplicate specimens archived at the Department of Botany, National Museums of Kenya, for future reference. The collected plant material was chopped into small pieces and spread on a wooden bench to dry naturally in a well-ventilated room for two weeks. Once dried, the material was ground into a powder using an electric plant grinder, and the resulting powder was packed and stored in a labelled khaki envelope pending extraction.

### 2.2 Extraction

The cold maceration method described previously [12] was employed following sight modifications [13]. Briefly, 100 grams of the powdered plant material were macerated in 1000 millili-tres of analytical grade ethyl acetate in a well-labelled conical flask for 48 hours, with occasional agitation using a mechanical shaker. The resulting mixture was decanted and filtered through Whatman No. 1 filter paper into a separate flask. And concentrated *in vacuo* at 50˚C using a rotary evaporator. The extract was further dried at 30˚C for five days in a hot air oven, and percentage yield was computed according to Eq 1 [14].

$$\% \text{ Yield} = \frac{\text{weight of the extract}}{\text{weight of the macerated sample}} \text{ x } 100 \tag{1}$$

### 2.3 Determination of *in vitro* antioxidant effects of the study plant extract

**2.3.1 Determination of 1,1-Diphenyl-2-picryl Hydrazyl Radical (DPPH) scavenging activity.** The experimental procedure was adapted from an established method [15] with slight modifications [13]. A 0.3 mM DPPH solution was prepared by dissolving 12 mg of 1,1-diphenyl-2-picrylhydrazyl in 100 ml of analytical-grade methanol. Test samples, including the extract and ascorbic acid, were prepared at concentrations of 1000, 100, 10, 1, 0.1, and 0.01 µg/ml. In triplicate, 1 ml of each test sample was mixed with 2.5 ml of the DPPH solution and incubated in the dark for 15 minutes. Absorbance was measured at 517 nm using a Shi-madzu UV-VIS 1600 spectrophotometer within 60 minutes, against a control. The percentage of DPPH radical scavenging activity (% RSA) was calculated using Eq 2, and the $EC_{50}$ values, indicating the concentration required to scavenge 50% of DPPH radicals, were determined

using linear regression plot of % RSA versus concentration to evaluate antioxidant activity.

$$\% \, \text{RSA} = \left[\frac{\text{Absorbance of control} - \text{Absorbance of test sample}}{\text{Absorbance of control}}\right] \times 100 \qquad (2)$$

**2.3.2 Hydrogen peroxide scavenging activity.** The hydrogen peroxide scavenging activity (HPSA) of the plant extract was evaluated using a modified protocol [16]. Briefly, 600 µl of 40 mM hydrogen peroxide in 0.1 M phosphate buffer (pH 7.4) was mixed with 1 ml of the plant extract or L-ascorbic acid across a concentration range of 0.01–1000 µg/ml. Following a 10-minute incubation at room temperature, absorbance was measured at 230 nm with a Shimadzu UV-Vis 1600 double-beam spectrophotometer, using phosphate buffer as the blank. The percentage HPSA was calculated according to Eq 3 [16], and the $EC_{50}$ value, representing the concentration required to scavenge 50% of hydrogen peroxide, was determined from a linear regression plot of % HPSA versus concentration.

$$\% \, \text{HPSA} = 1 - \left[\frac{\text{Absorbance of control} - \text{Absorbance of test sample}}{\text{Absorbance of control}}\right] \times 100 \qquad (3)$$

**2.3.3 Determination of total antioxidant capacity.** In this experiment, we employed a modified phosphomolybdenum method [17] to assess the extract's total antioxidant capacity. L-ascorbic acid concentrations (0.01 to 1000 µg/ml) was prepared and added in triplicate to a reaction mixture containing 0.6 M sulfuric acid, 28 mM sodium phosphate, and 4 mM ammonium molybdate. The samples were incubated at 95°C for 90 minutes, and cooled to room temperature, and their absorbances were recorded at 695 nm using a Shimadzu UV-Vis 1600 double-beam spectrophotometer, with 300 µl methanol serving as the blank. A linear regression curve of absorbance versus L-ascorbic acid concentration was generated to interpolate the ascorbic acid concentrations in plant extracts. Each plant extract (300 µl) was treated identically, with absorbance measured at 695 nm. All assays were performed in triplicate. The total antioxidant capacity of the extract was quantified as ascorbic acid equivalent per milligram of dry weight (µg AAE/mg d.w), providing a measure of antioxidant capacity.

## 2.4 Determination of *in vitro* cytotoxic effects of the study extract against normal mammalian and PCa cells

**2.4.1 Preparation of plant extracts.** The plant extract was precisely weighed (100 mg), dissolved in dimethyl sulphoxide (DMSO) (10 ml), and thoroughly vortexed. Subsequently, 100 µl of the solubilized extract was diluted with phosphate-buffered saline (PBS) to a final volume of 1000 µl, achieving a stock concentration of 1000 µg/ml. Serial dilutions using a three-fold factor produced seven concentrations: 1.37 µg/ml, 4.12 µg/ml, 12.35 µg/ml, 37.04 µg/ml, 111.11 µg/ml, 333.33 µg/ml, and 1000 µg/ml of the extract. The reference drug, Doxorubicin, was similarly prepared, yielding concentrations of 0.04 µg/ml, 0.12 µg/ml, 0.37 µg/ml, 1.11 µg/ml, 3.33 µg/ml, 10 µg/ml, and 30 µg/ml.

**2.4.2 Cell culture and maintenance.** The cell lines were cultured and maintained using a standard protocol [18]. Vero CCL-81 cells were grown in Eagle's Minimum Essential Medium (EMEM) supplemented with 10% foetal bovine serum (FBS), 2 mM L-glutamine, and antibiotics (100 U/ml penicillin and 0.1 mg/ml streptomycin). DU-145 cells were cultured in high-glucose Dulbecco's Modified Eagle Medium (DMEM) with similar supplements. All cultures were incubated at 37°C in a humidified incubator supplied with 5% $CO_2$ for 48 hours. Cells

were routinely monitored, trypsinised, and passaged, and those at $\geq$90% confluence were harvested and prepared for subsequent assays.

**2.4.3 3-(4,5-dimethylthiazol-2-yl)-2,5-diphenyltetrazolium bromide (MTT) assay for cytotoxicity and cancer cell inhibition.** The cytotoxicity of plant extract on normal mammalian cells (Vero CCL-81) was evaluated using the MTT assay [19] with slight modifications [18]. Cells were seeded at a density of $2\times10^4$ cells/well in 96-well plates with 100 μL of medium. Following overnight incubation at 37˚C with 5% $CO_2$, the cells were treated in triplicate with serial dilutions of plant extracts (0–1000 μg/mL in 0.1% DMSO) and the reference drug, Doxorubicin. After 48 hours of incubation under the same conditions, 10 μL of freshly prepared MTT reagent (5 mg/mL in PBS) was added to each well, and plates were incubated for an additional 4 hours, as detailed in section 2.4.2. Supernatants were aspirated, replaced with 100 μL of DMSO (0.1%), and agitated. Then, absorbance was measured at 570 nm and used to calculate cytotoxicity/inhibition percentages according to Eq 4.

$$\% \text{ Cytotoxicity inhibition} = 1 - \left(\frac{\text{The absorbance of treated cells}}{\text{The absorbance of control cells}}\right) \times 100 \qquad (4)$$

Furthermore, the median cytotoxic concentrations ($CC_{50}$) for Vero CCL-81 cells and median inhibitory concentrations ($IC_{50}$) for DU-145 cells were derived from a linear regression plot of percentage cytotoxicity/inhibition versus concentration to evaluate the safety of the extract. Moreover, the selectivity index (SI) of the plant extract was calculated using Eq 5, described previously [20].

$$\text{SI} = \left(\frac{CC_{50} \text{ value for Vero CCL} - 81 \text{ cells}}{IC_{50} \text{ value for cancer cells}}\right) \qquad (5)$$

**2.4.4 Determination of the expression profiles of cancer-associated genes in the DU-145 cells treated with the selected plant extract.** The expression profiles of key genes involved in cancer initiation and progression were analysed using quantitative real-time PCR (qPCR). DU-145 cells ($1\times10^6$) were seeded in 96-well plates and treated with the plant extract at a concentration corresponding to its $IC_{50}$ and upon the determination of its selectivity, based on SI value, followed by a 48-hour incubation at conditions described in section 2.4.2. Total RNA was extracted using a miniprep kit (Solis BioDyne) and quantified with a nanodrop spectrophotometer (ThermoFisher Scientific). RNA (2 μg) was reverse transcribed into cDNA using a cDNA synthesis kit (Solis BioDyne), added SYBR Green dye, and then amplified using specific primers (Table 1) for *Bcl2*, *ar*, *cdk1*, *p53*, *caspase 3*, with *actb* and *gapdh* as the normalisation controls. Gene expression was measured with a QuantStudio™ 5 System (ThermoFisher Scientific) and analysed using the comparative threshold (CT) method, with fold changes calculated via the $2^{-\Delta\Delta Ct}$ approach [21].

## 2.5 Phytochemical analysis

Phytochemical analysis was conducted using gas chromatography-mass spectrometry (GC-MS), following a standard protocol [22]. Analytical grade ethyl acetate (Sigma-Aldrich) was used. Standard reference compounds for GC-MS calibration and comparison were also obtained from Sigma-Aldrich. The ethyl acetate stem bark extract of *B. coriacea* was dissolved using its own solvent to get a concentration of 1 mg/ml. The sample solution was filtered through a 0.45μm PTFE syringe filter, then injected into a Shimadzu QP 2010-SE GC-MS equipped with an auto-sampler. Ultrapure helium (He) served as the carrier gas at a linear velocity of 35 cm/s. Separation was achieved using a BPX5 nonpolar column (30 m × 0.25 mm

**Table 1. Target genes and their respective primers.**

| Target gene | | Primer sequence | |
|---|---|---|---|
| | | Forward [5'-3'] | Reverse (3'-5') |
| 1. | *actb* | GCCAACTTGTCCTTACCCAGA | AGGAACAGAGACCTGACCCC |
| 2. | *p53* | CTTCGAGATGTTCCGAGAGC | GACCATGAAGGCAGGATGAG |
| 3. | *Caspase3* | CAAAGAGGAAGCACCAGAACCC | GGACTTGGGAAGCATAAGCGA |
| 4. | *cdk1* | GAACACCACTTGTCCCTCTAAGAT | CTGCTTAGTTCAGAGAAAAGTGC |
| 5. | *bcl2* | GGCCTCAGGGAACAGAATGAT | TCCTGTTGCTTTCGTTTCTTTC |
| 6. | *ar* | GCTTTATCAGGGAGAACAGCCT | TGCAGCTCTCTCGCAATCTG |
| 7. | *gapdh* | CCCCACCACACTGAATCTCC | CTCACCTTGACACAAGCCCA |

ID; 0.25 μm film thickness). The GC oven was programmed from 60˚C, ramping at 10˚C/min to 200˚C (hold for 1 min), then at 10˚C/min to 280˚C (hold for 10 min), totalling a runtime of 33 minutes. The injection temperature was 250˚C, the interface temperature was 250˚C, and the EI ion source operated at 200˚C in electron impact (EI) mode at 70 eV. Mass spectrometry was conducted in Scan mode over an m/z range of 35–550 a.m.u., with sample injection in split mode at a split ratio of 10:1.

## 2.6 Data management, statistical analysis, and reporting

*In vitro* antioxidant, cytotoxicity, and antiproliferative data were initially organized in Microsoft Excel (Microsoft 365) and subsequently analysed using Minitab software version 21.4. Descriptive statistics were computed, and results were expressed as mean ± standard deviation ($\bar{x} \pm SD$) from replicate experiments. Normality of the quantitative data from antioxidant and cytotoxicity experiments was assessed using the Shapiro-Wilk test, and deemed normally distributed. Afterward, statistical comparisons among means were conducted using unpaired student t-test or One-Way Analysis of Variance (ANOVA) with Tukey's *post hoc* test at a significance level of $\alpha = 0.001$, and graphical presentation of the findings was performed using GraphPad Prism version 10 software. Gene expression data were analysed using the $2^{-\Delta\Delta Ct}$ method via the QuantStudio™ 5 System's software. Compounds were identified by matching mass spectra against the NIST library and calculating retention indices with a homologous series of n-alkanes (C8-C20) under the same GC-MS conditions. Verification involved comparing mass spectra and retention indices with standards on the National Library of Medicine's PubChem (nih.gov), NIST Chemistry WebBook, and scholarly literature. The identified compounds were catalogued by name, class, molecular weight, molecular formula, and retention time, with their abundance determined from peak areas in the total ion chromatogram. Agilent ChemStation software was used for data analysis.

## 2.7 Ethics approval and consent to participate

The present study and all its protocols were approved by the Kenyatta University Ethics Review Committee (reference number: PKU/2652/11787) and the National Commission for Science, Technology, and Innovation (reference number: NACOSTI/P/23/25162). Consent to participate was not required for this study.

## 3. Results

### 3.1 *In vitro* antioxidant activity

**3.1.1 DPPH radical scavenging activity of stem bark extracts of *B. coriacea*.** The evaluation of DPPH radical scavenging activity of the ethyl acetate stem bark extract of *B. coriacea*

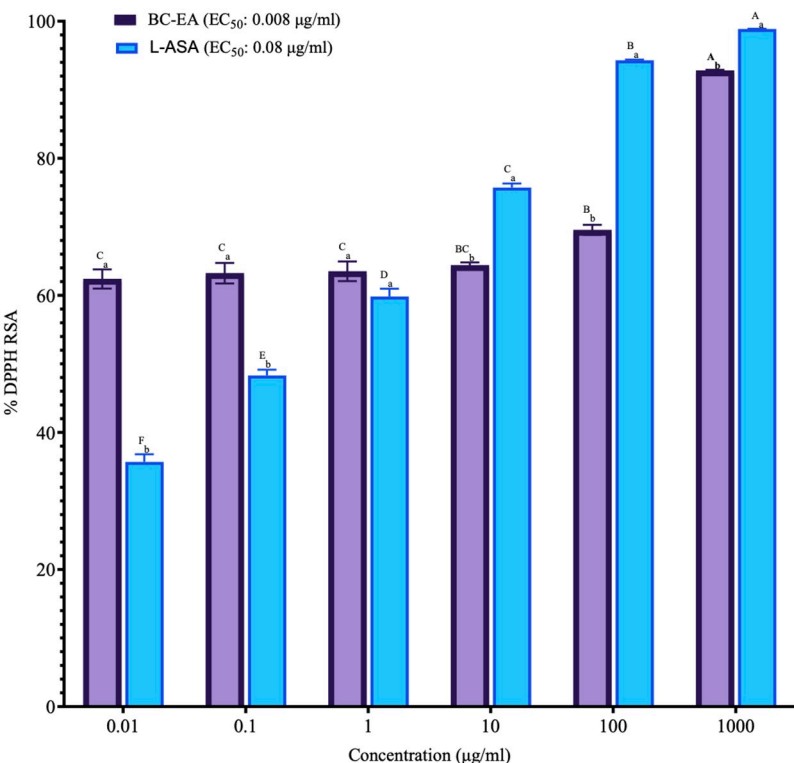

**Fig 1. *In vitro* DPPH radical scavenging effects of ethyl acetate stem bark extract of *B. coriacea*.** The results are presented as $\bar{x} \pm SD$ for triplicate experiments. Bars with different subscript lowercase letters within the same concentration are significantly different (P<0.0001; Unpaired student t-test) and those with different uppercase superscript letters across concentrations are significantly different (P<0.001; One-Way ANOVA with Tukey's *post hoc* test); % DPPH RSA: Percentage 1,1-Diphenyl-2-picryl Hydrazyl Radical Scavenging Activity; BC-EA: Ethyl acetate stem bark extract of *B. coriacea*; L-ASA: L-Ascorbic Acid.

revealed no significant differences at concentrations of 0.01 μg/ml, 0.1 μg/ml, and 1 μg/ml, as well as at 10 μg/ml and 100 μg/ml (P>0.001; Fig 1; S1 File). However, a significant increase in percentage scavenging activity was observed with increasing concentrations, (P<0.001; Fig 1; S1 File). Besides at concentrations of 0.01 μg/ml and 0.1 μg/ml, the ethyl acetate stem bark extract of *B. coriacea* showed significantly higher DPPH radical scavenging activities than those of L-ascorbic acid (P<0.001; Fig 1; S1 File). At a concentration of 1 μg/ml, no significant difference between the DPPH radical scavenging activity of the studied plant extract and L-ascorbic acid (P>0.001; Fig 1; S1 File). However, at concentrations of 10 μg/ml, 100 μg/ml, and 1000 μg/ml, L-ascorbic acid exhibited significantly higher DPPH radical scavenging activities than the ethyl acetate stem bark extract of *B. coriacea* (P<0.001; Fig 1; S1 File).

Furthermore, it was observed that the $EC_{50}$ value of the study plant extract (0.008 μg/ml) was tenfold lower compared to hat L-ascorbic acid (0.08 μg/ml) (Fig 1).

**3.1.2 Hydrogen peroxide scavenging activity of the stem bark extracts of *B. coriacea*.**
The ethyl acetate stem bark extract of *B. coriacea* displayed no significant differences in hydrogen peroxide scavenging activity at concentrations of 1 μg/ml to 1000 μg/ml (P > 0.001), though these activities were markedly higher than those observed at 0.01 μg/ml and 0.1 μg/ml (P < 0.001; Fig 2; S2 File). Likewise, the hydrogen peroxide scavenging activities of the ethyl acetate stem bark extract of *B. coriacea* at concentrations of 0.1 μg/ml to 10 μg/ml were not significantly different (P>0.001; Fig 2; S2 File). Notably, the hydrogen peroxide scavenging

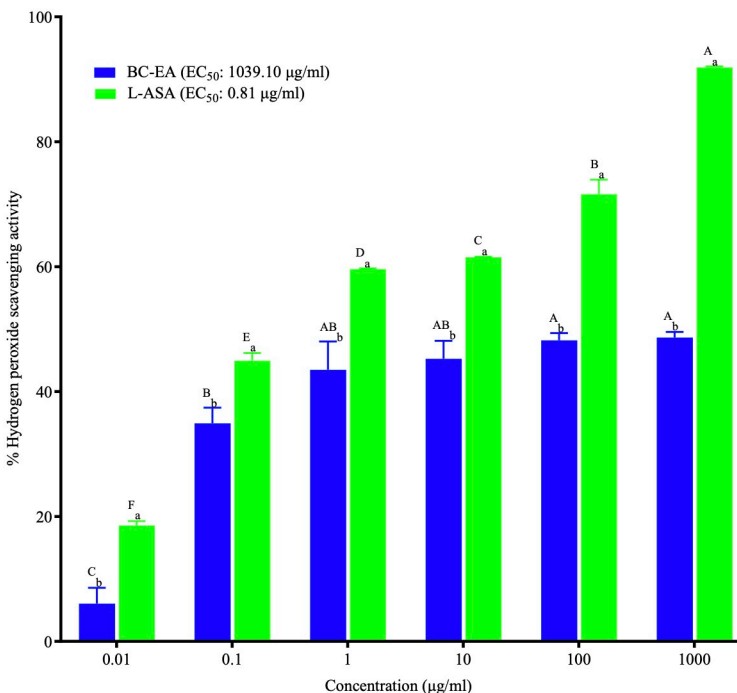

**Fig 2. Hydrogen peroxide scavenging activity of the ethyl acetate stem bark extract of *B. coriacea*.** The results are presented as $\bar{x} \pm SD$ for triplicate experiments. Bars with different subscript lowercase letters within the same concentration are significantly different (P<0.0001; Unpaired student t-test) and those with different uppercase superscript letters across concentrations are significantly different (P<0.001; One-Way ANOVA with Tukey's *post hoc* test); BC-EA: Ethyl acetate stem bark extract of *B. coriacea*; L-ASA: L-Ascorbic Acid.

activity of L-ascorbic acid increased significantly with increasing concentration (P<0.001; Fig 2; S2 File).

At all the tested concentrations, the ethyl acetate stem bark extract of *B. coriacea* showed significantly lower hydrogen peroxide scavenging activity than L-ascorbic acid (P<0.001; Fig 2; S2 File). Additionally, the ethyl acetate extract had a considerably higher $EC_{50}$ (1039.10 μg/ml) than L-ascorbic acid (Fig 2).

**3.1.3 Total antioxidant capacity of the stem bark extracts of *B. coriacea*.** This study showed that the total antioxidant capacity of the ethyl acetate stem bark extract of *B. coriacea* was high at 454.39±25.26 μg AAE /mg dw.

## 3.2 *In vitro* cytotoxicity and antiproliferative effects

**3.2.1 *In vitro* cytotoxicity of the ethyl acetate stem bark extract of *B. coriacea*.** Overall, the ethyl acetate stem bark extract of *B. coriacea* exhibited positive significant (P<0.001) concentration-dependent cytotoxicity against Vero CCL-81 cells (Fig 3; S3 File). No significant difference between the cytotoxicity of the studied plant extract at concentrations of 333.33 μg/ml and 1000 μg/ml, and among those observed at concentrations of 12.35 μg/ml, 37.04 μg/ml, and 111.11 μg/ml, respectively (P>0.001; Fig 3; S3 File). Besides, the cytotoxicity of doxorubicin at concentrations of 3.33 μg/ml, 10.00 μg/ml, and 30.00 μg/ml (P>0.001), between 1.11 μg/ml and 3.33 μg/ml (P>0.001), 0.12 μg/ml and 0.37 μg/ml (P>0.001), and 0.04 μg/ml and 0.12 μg/ml (P>0.001), respectively, was not significantly different (Fig 3; S3 File). Furthermore, the $CC_{50}$ of the ethyl acetate extract was 68.61 μg/ml, and higher than that of doxorubicin (0.41 μg/ml) as shown in Fig 3.

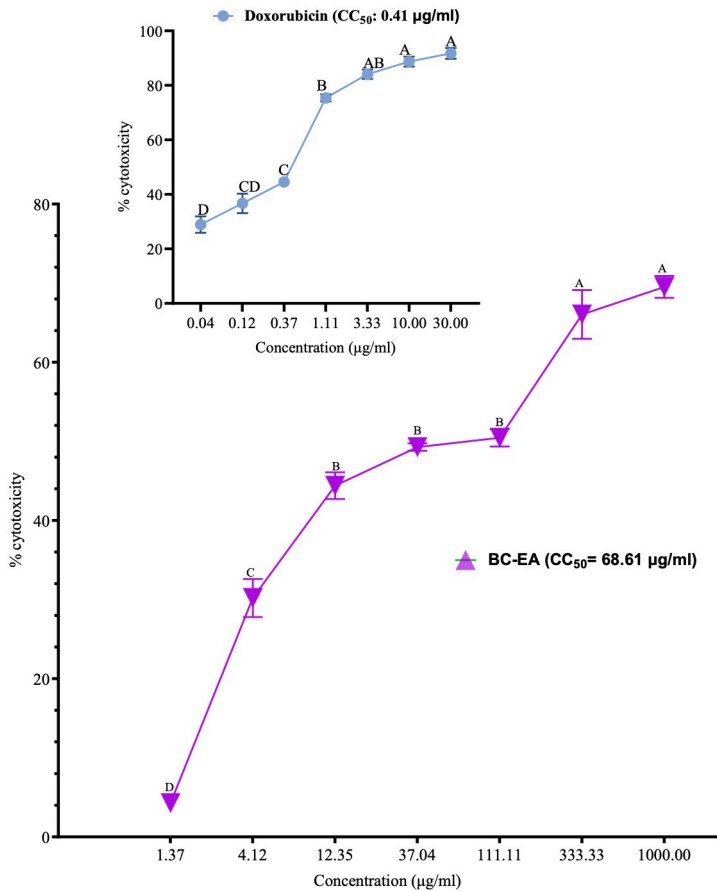

**Fig 3. Cytotoxic effects of the ethyl acetate stem bark extract of *B. coriacea* against Vero CCL-81 cells.** The results are presented as $\bar{x} \pm SD$ for four triplicate experiments. Different uppercase letters across concentrations are significantly different (P<0.001; One-Way ANOVA with Tukey's *post hoc* test); BC-EA: Ethyl acetate stem bark extract of *B. coriacea*; $CC_{50}$: Median cytotoxic concentration.

**3.2.2 Effects of the ethyl acetate stem bark extract of *B. coriacea* on DU-145 cells.** The inhibitory effects of the ethyl acetate stem bark extracts of *B. coriacea* against DU-145 cells did not differ significantly at concentrations of between 1.37 and 12.35 μg/ml and between 111.11 and 333.33 μg/ml (P>0.001; Fig 4; S4 File). Also, the difference between the inhibitory effects of the studied plant extract at concentrations of between 37.04 μg/ml and 111.11 μg/ml were insignificant (P>0.001; Fig 4; S4 File). Nevertheless, the percentage inhibition of DU-145 by this extract increased significantly with increasing extract concentration (P<0.001; Fig 4; S4 File). Besides, no significant difference was observed between the inhibition of DU-145 cells by doxorubicin at concentrations of 10 μg/ml and 30 μg/ml, was observed (P>0.001; Fig 4; S4 File); however, the inhibitions recorded at lower concentrations increased significantly with increasing concentration (P<0.001; Fig 4; S4 File). Moreover, the $IC_{50}$ of this extract was 32.16 μg/ml and higher than that of the reference drug, doxorubicin (0.28 μg/ml) (Fig 4).

**3.2.3 Selectivity index.** The selectivity index (SI) of the studied plant extract was computed to determine its ability to selectively exert cytotoxic effects on cancer cells while sparing normal cells. The results showed that the ethyl acetate stem bark extract of *B. coriacea* was 2.13 and relatively higher than that of doxorubicin (1.46).

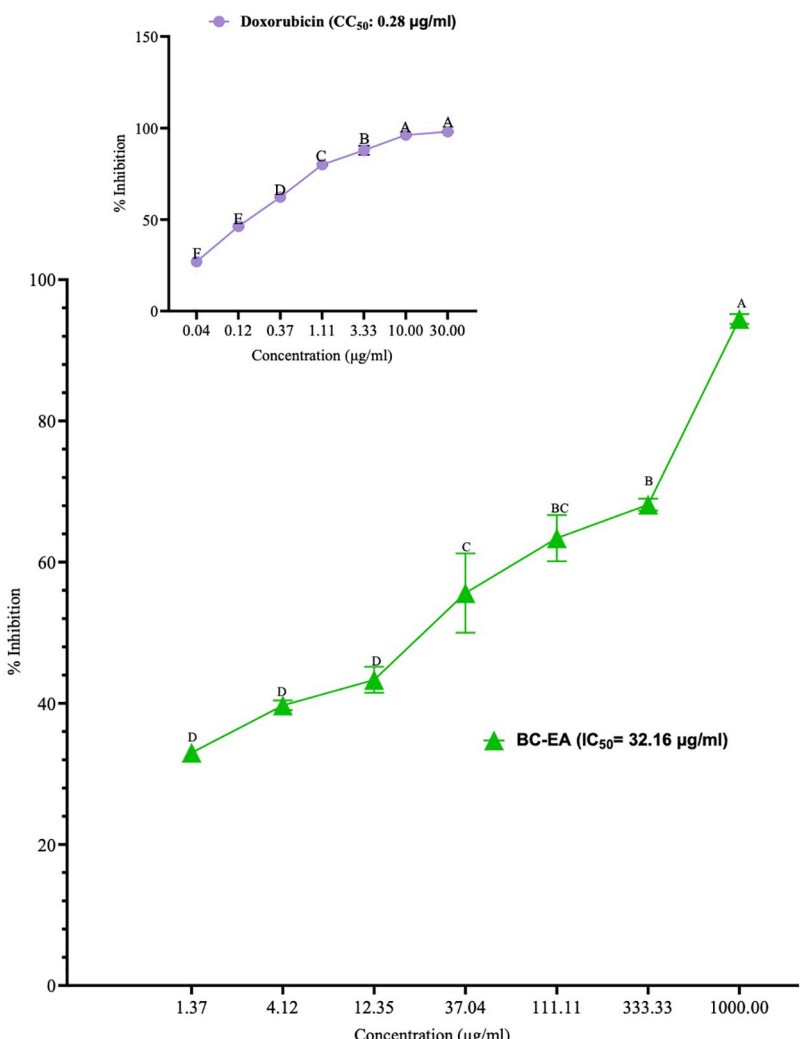

**Fig 4. Anti-PCa effects of the ethyl acetate stem bark extract of *B. coriacea* against DU-145 cells.** The results are presented as $\bar{x} \pm SD$ for four triplicate experiments. Different uppercase letters across concentrations are significantly different (P<0.001; One-Way ANOVA with Tukey's *post hoc* test); BC-EA: Ethyl acetate stem bark extract of *B. coriacea*; $CC_{50}$: Median cytotoxic concentration.

### 3.3 Expression levels of selected genes in DU-145 cells treated with the selected plant extracts

It was also observed that the DU-145 cells treated with the ethyl acetate stem bark extract of *B. coriacea* showed a significantly lower expression of the *ar*, *bcl₂*, *caspase 3* and *cdk1* genes (P<0.001; Table 2). Notably, the expression of *p53* gene was significantly higher in DU-145 treated with the studied plant extract than in the control cells (P<0.001; Table 2).

### 3.4 Phytochemical compounds detected in the ethyl acetate stem bark extract of *B. coriacea*

Phytochemical analysis of the ethyl acetate (BC-EA) showed the presence of 19 compounds distributed in various groups of chemical classes including glyceride esters, glycerides, hydantoins, phenolic aldehydes, alkenes, phthalates, aliphatic alcohol fatty acids, esters, diterpenoids,

**Table 2. Target gene expression profile in DU-145 cells treated with the ethyl acetate stem bark extract of *B. coriacea*.**

| Target gene | Expression Level (ROQ values) | |
| --- | --- | --- |
| | BC-EA | Ctrl |
| *Ar* | $0.046 \pm 0.000^b$ | $1.000 \pm 0.000^a$ |
| *bcl$_2$* | $0.006 \pm 0.000^b$ | $1.000 \pm 0.000^a$ |
| *caspase 3* | $0.470 \pm 0.000^b$ | $1.000 \pm 0.000^a$ |
| *cdk-1* | $0.006 \pm 0.000^b$ | $1.000 \pm 0.000^a$ |
| *p$^{53}$* | $2.129 \pm 0.000^a$ | $1.000 \pm 0.000^b$ |

The results are presented as $\bar{x} \pm SD$ for three replicate experiments. Means with different superscript letters within the same column are significantly different ($P < 0.001$; Unpaired student t-test); BC-EA: Ethyl acetate stem bark extract of *B. coriacea* (32.16 µg/ml); Ctrl: Control/Untreated; *ar*: Androgen receptor gene; *bcl$_2$*: B-cell lymphoma 2 gene; *caspase 3*: Cysteine-aspartic acid protease gene; *cdk-1*: Cyclin-dependent Kinase 1/ Cell division cycle protein 2 gene; *p$^{53}$*: Tumour Protein 53 gene.

as well as aliphatic amine, alkylated alkanes, and phytosterols. Notably, Lup-20(29)-en-3-one (7.99%) and Lupeol (59.49%) were the most dominant (Table 3). Conversely, 4-Hydroxy-2-methoxybenaldehyde (0.22%) and 1-Octadecene (2.50%) were the least abundant compounds in this extract (Table 3). The obtained chromatograph is presented in Fig 5.

## 4. Discussion

The antioxidant research landscape has undergone a significant transformation in recent years, with a growing emphasis on identifying natural compounds that can mitigate oxidative stress and its associated disorders, including cancer, diabetes, and neurodegenerative disorders

**Table 3. Phytochemicals identified in the ethyl acetate stem bark extract of *B. coriacea*.**

| Compound class | Compound name | Molecular Formula | Molecular weight | Retention Time (Min) | % Area |
| --- | --- | --- | --- | --- | --- |
| Glyceride ester | 1,2,3-Propanetriol, 1-acetate | $C_5H_{10}O_4$ | 134 | 7.051 | 1.96 |
| Glyceride | Glycerol α-monoacetate | $C_5H_{10}O_4$ | 134 | 9.266 | 1.86 |
| Hydantoin | 5-Ethylhydantoin | $C_5H_8N_2O_2$ | 128 | 10.595 | 3.35 |
| Phenolic aldehyde | 4-Hydroxy-2-methoxybenaldehyde | $C_8H_8O_3$ | 152 | 11.802 | 0.22 |
| Phenolic | 2,4-Di-tert-butylphenol | $C_{14}H_{22}O$ | 206 | 12.995 | 1.91 |
| Alkene | 9-Eicosene, (E)- | $C_{20}H_{40}$ | 280 | 13.878 | 2.47 |
| Alkene | 1-Octadecene | $C_{18}H_{36}$ | 252 | 16.249 | 2.50 |
| Phthalate | 1,2-Benzenedicarboxylic acid, bis(2-methylpropyl) ester | $C_{16}H_{22}O_4$ | 278 | 17.248 | 1.46 |
| Aliphatic alcohol | 1-Heneicosanol | $C_{21}H_{44}O$ | 312 | 18.387 | 2.57 |
| Fatty acid ester | Ethyl tridecanoate | $C_{15}H_{30}O_2$ | 242 | 18.434 | 0.71 |
| Diterpenoid | Neophytadiene | $C_{20}H_{38}$ | 278 | 19.634 | 1.03 |
| Aliphatic amine | N-Ethyldiethanolamine, | $C_{12}H_{31}NO_2Si_2$ | 277 | 19.846 | 1.17 |
| Aliphatic alcohol | n-Nonadecanol-1 | $C_{19}H_{40}O$ | 284 | 20.336 | 1.64 |
| Fatty acid ester | 4-Hexenoic acid, 3-hydroxy-2-methyl-, methyl ester, (R*, R*)- | $C_8H_{14}O_3$ | 158 | 20.370 | 0.50 |
| Alkane | Eicosane, 2-methyl- | $C_{21}H_{44}$ | 296 | 21.269 | 0.24 |
| Fatty acid ester | Heptadecyl trifluoroacetate | $C_{19}H_{35}F_3O_2$ | 352 | 22.113 | 1.18 |
| Phytosterol | γ-Sitosterol | $C_{29}H_{50}O$ | 414 | 25.896 | 7.77 |
| Triterpenoid | Lup-20(29)-en-3-one | $C_{30}H_{48}O$ | 424 | 28.634 | 7.99 |
| Triterpenoid | Lupeol | $C_{30}H_{50}O$ | 426 | 29.960 | 59.49 |

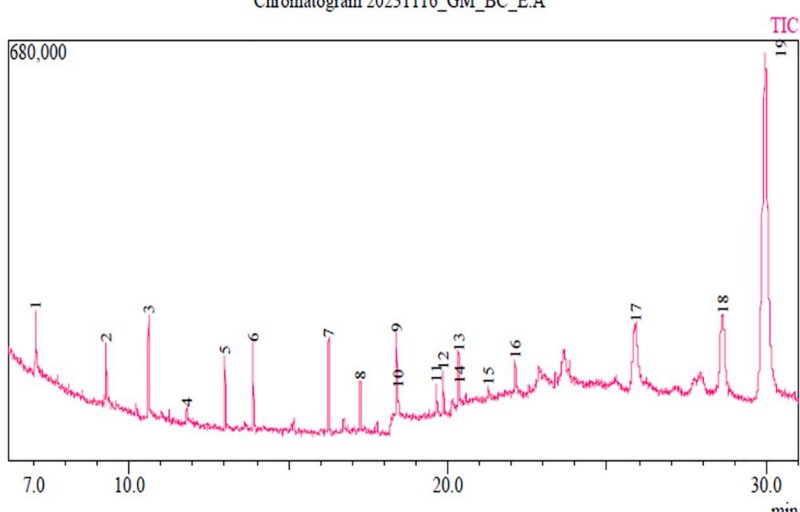

**Fig 5. A GC-MS chromatogram showing peaks of the detected phytochemical compounds in the ethyl acetate stem bark extract of *B. coriacea*.**

[23]. Synthetic antioxidants, such as BHA and BHT, cause undesirable effects, including cancer and autoimmunity, limiting their clinical significance. Contrastingly, natural antioxidants derived from medicinal plants are considered safer and more effective [24]. Therefore, the antioxidant potential of the ethyl acetate stem bark extract of *B. coriacea* was investigated to appraise its potential in averting OS and associated complications, especially cancer of the prostate.

The DPPH assay technique is reputed for its accuracy and reproducibility in measuring the ability of a compound to scavenge DPPH radicals, leading to a colour change from purple/violet to yellow, depending on the concentration of the antioxidant, and a reduction in absorbance measured at 517 nm [25]. The results showed that the ethyl acetate stem bark extract of *B. coriacea* exhibited significant DPPH radical scavenging activity, with a concentration-dependent increase in efficacy. This suggests that the extract contains robust antioxidants, emphasizing its potential as a putative source of antioxidant compounds [26].

The hydrogen peroxide scavenging activity of the ethyl acetate stem bark extract of *B. coriacea* was also investigated using a previously described procedure [27]. Hydrogen peroxide is a significant ROS that can penetrate cells and induce DNA strand breaks, leading to genetic instability and various disorders, including cancer [28]. The results showed that the studied lant extract possesses considerable hydrogen peroxide scavenging efficacy, suggesting its potential in preventing the generation of hydroxyl radicals and their devastating effects on the cell.

The total antioxidant capacity (TAC) of the ethyl acetate stem bark extract of *B. coriacea* was evaluated using a previously established method [17]. The results showed that the extract possessed significant TAC, indicating its potential in preventing and alleviating oxidative stress [29]. The observed TAC further typifies the various antioxidant effects reported in this study, underscoring the potential of the studied plant extract as a promising source of potent antioxidant compounds which can be optimized to avert oxidative stress.

The quest for effective and safe anticancer therapies remains a pressing concern, particularly in less-developed countries where cancer burden is disproportionately high (~ 80%) exacerbated by limited healthcare resources [4]. In this context, the exploration of plant-derived

compounds as potential sources of novel anticancer agents is a feasible and viable alternative strategem to identify, optimise, and develop efficacious, safe, and cost-effective armamentaria. The cytotoxic and anti-PCa effects of the ethyl acetate stem bark extract of *B. coriacea* (Pax.) were investigated in this study to appraise its efficacy.

The *in vitro* cytotoxic effects of the studied plant extract on normal mammalian cells (Vero CCL-81) and cancer cell line (DU-145) were investigated using the 3-(4,5-dimethylthiazol-2-yl)-2,5-diphenyltetrazolium bromide (MTT) assay technique [18]. This method is renowned for its high sensitivity, reliability, and reproducibility, making it an ideal tool for assessing cell viability in drug toxicity and anticancer efficacy studies [30]. In the present investigation, the ethyl acetate stem bark extract of *B. coriacea* exhibited concentration-dependent cytotoxicity with moderate efficacy (30 μg/ml < $CC_{50}$ < 100 μg/ml) towards Vero CCL-81 cells, according to the criteria stipulated by the National Cancer Institute [31]. This concentration-dependent increase in cytotoxicity can be attributed to the increasing concentration of certain toxic phytochemicals [32].

The results further demonstrated significant extract-dependent increases in percentage inhibitions of growth of cancer cells, with the ethyl acetate stem bark extract of *B. coriacea* showing remarkable antiproliferative efficacy ($CC_{50}$ < 30 μg/ml). This finding is consistent with previous reports indicating concentration-dependent antiproliferative effects of plant extracts on cell lines [33]. The observed antiproliferative effects of the ethyl acetate stem bark extract of *B. coriacea* can be attributed to the presence of bioactive phytochemicals, such as phenolics, flavonoids, alkaloids, some types of phytosterols and terpenoids. These phytochemicals have been shown to exert anticancer activity against various cancers through various mechanisms [34].

Selectivity index (SI) is a robust parameter to deduce the ability of plant extracts and chemicals to inhibit the proliferation of cancerous cells while selectively sparing normal/health cells [35]. In this study, the ethyl acetate stem bark extract of *B. coriacea* exhibited high selectivity against DU-145 cells, compared to Doxorubicin's moderate electivity, underscoring the presence of effective phytochemicals which can target unique processes of cancerous cells to inhibit their proliferation and survival. Additionally, the presence of antioxidant compounds in this extract, certainly thwarted free radical-mediated oxidative damage to essential biological molecules in healthy mammalian cells, while perpetuating damage to cancerous cells as reported previously [36].

The primary mechanism of action attributed to doxorubicin involves its inherent capability to intercalate within DNA pairs, thereby instigating the fragmentation of DNA strands and impeding DNA and RNA synthesis [37]. Additionally, doxorubicin precipitates free radical-mediated oxidative damage to DNA in conjunction with iron, imposing supplementary restrictions on DNA synthesis [38]. However, the lower selectivity of doxorubicin against cancerous cells is associated with adverse effects, such as fatigue, alopecia, nausea, vomiting, and oral sores, along with notable instances of bone marrow suppression and elevated susceptibility to secondary malignancies, among other devastating complications [38, 39]. Thus, considering the drawbacks bedevilling doxorubicin therapy and other conventional chemotherapeutic agents, the utilization of the promising plant extracts and their valorisation based on empirical evidence, such as this study, is a viable alternative to mitigating the adverse effects. Notably, the high selectivity of the ethyl acetate stem bark extract of *B. coriacea* can be linked to the presence of phytochemicals which induced DNA and RNA damage in the DU-145 cells, while sparing health cells (Vero CCL-81), underpinning its anticancer efficacy and safety. However, further empirical investigations, using other cell lines and *in vivo* models are required to clarify the promising effects reported herein, and to facilitate the development of safe and effective botanically-derived anticancer agents.

The progression of PCa is a complex process involving the dysregulation of critical genes governing the cell cycle, proto-oncogenes, and tumour-suppressing genes [40]. The androgen receptor (AR) plays a crucial role in PCa progression [41]. When activated through its ligand binding domain (LBD), the AR translocates into the nucleus and forms dimers that bind to the androgen response element (ARE) on target genes [42], recruiting transcriptional machinery, and facilitating gene transcription [43]. Pharmacological compounds inhibiting AR action target either the ligand binding domain (LBD) of the androgen receptor gene (*ar*) or interactions with co-regulatory proteins, impeding disease progression [44]. It was observed that the expression of the *ar* gene in PCa cells (DU-145) treated with the ethyl acetate stem bark extract of *B. coriacea* was significantly lower than in the control cells. These findings imply that the antiproliferative effects of this extract observed in this study are partly driven through the inhibition of the *ar* gene overexpression [45].

The B-Cell Lymphoma-2 (BCL-2) family of proteins comprise anti-apoptotic proteins (BCL-2, BCL-XL, MCL-1, BCL-W, and BFL-1) and pro-apoptotic proteins (BAK, BAX, BOK and BH3-only proteins). The pro-apoptotic proteins are categorised as sensitizers (NOXA, BAD, HRK, BMF, and BIK) and activators (BID, BIM, and PUMA) [46]. The intricate interplay between pro- and anti-apoptotic protein levels determines a cell's fate- whether it lives or dies [47]. The expression of the *bcl-2* gene was significantly lower in DU-145 cells treated with the ethyl acetate stem bark extract of *B. coriacea* than in the control cells. This finding underscores the potential of the studied plant extract potential in averting the *bcl-2* overexpression in PCa cells to promote their apoptosis.

The Caspase family represents a group of proteins, homologous to the *Caenorhabditis elegans'* cell death abnormal-3 gene (CED-3) [48]. This protein family is distinctly composed of initiator caspases (caspase-2, -8, -9, -10), executioner caspases (caspase-3, -6, -7), and inflammatory caspases (caspase-1, -4, -5, -11) [49]. Depletion of caspase activity, especially caspase-3, plays a crucial role in tumour progression, as it occupies a terminal position in the caspase cascade, activated by both intrinsic and extrinsic death pathways in apoptosis [50]. Previous research shows that the overexpression of *caspase 3* may reduce cancer patients' overall survival rate and confer tumour cell regrowth ability, chemotherapy immunity, and other adverse effects [51]. Notably, the expression of caspase-3 in DU-145 cells treated with the studied plant extract was normal, denoting the extract's ability to normalise its expression level to promote the execution of apoptosis [50].

Cyclin-dependent kinases (CDKs) are serine/threonine kinases that form complexes with cyclin proteins, a crucial process for fully activating their kinase activity [52]. Their pivotal roles extend to controlling cell division and transcription modulation in response to extracellular and intracellular stimuli [53]. Analysis from The Cancer Genome Atlas (TCGA) UAL-CAN database reveals significant upregulation of CDKs in cancerous tissues compared to normal tissues [54]. This study's findings depicted a normal expression of CDK1 in DU-145 cells treated with the ethyl acetate stem bark extract of *B. coriacea*. These results underscore this extract's ability to restore normal expression of CDK1 and probably its normal function in regulating the cell cycle, thus averting PCa. Further investigation of this extract's modulatory role on the cell cycle using other *in vitro* and *in vivo* models may provide crucial insights into its anticancer potential.

The *p53* gene encodes the p53 tumour suppressor protein, commonly known as the "Guardian of the Genome," which safeguards the integrity of cellular DNA and plays pivotal roles in development, ageing, and cell differentiation [55]. Genetic models with p53-nullizygous traits display phenotypes related to ageing, pluripotency, and development, characterised by early onset of ageing, induction of cell pluripotency, and the inability of embryos to undergo gastrulation [56]. Recent studies have revealed *p53*'s involvement in pathways such as

autophagy, cell metabolism, ferroptosis, and those generating reactive oxygen species [56]. The significantly higher expression of *p53* in DU-145 cells treated with the ethyl acetate stem bark extract of *B. coriacea* indicates the presence of key secondary metabolites which not only reactivate the expression of this gene but also increase its expression levels to promote apoptosis in prostate cancer cells [57]. Furthermore, determining the intricate mechanisms through which the studied plant extract influence the interplay of other genes may unravel the appropriate targets and facilitate the development of efficacious and safe anti-PCa drugs from this botanical resource.

Phytochemical analysis of the ethyl acetate stem bark extract of *B. coriacea* revealed a wide range of compounds, including triterpenoids, phenolics, and fatty acid esters. For instance, triterpenoids like lup-20(29)-en-3-one and lupeol, detected in this extract possess various biological effects, including anti-inflammatory, antioxidant, and anticancer activities [58]. Existing evidence shows these compounds prevent the development of cancer, by promoting apoptosis and inhibiting cancer cell proliferation and metastasis [58]. Lupeol exerts its anticancer effects by modulating key signalling pathways such as PI3K/Akt, MAPK/ERK, and JAK/STAT, and regulating the function of proteins driving cell cycle progression or apoptosis [58]. Besides, the high antioxidant and antiproliferative effects of the studied plant extract can be attributed to the high concentration of these phytochemicals. Research has demonstrated that phytosterols promote proper immune response functioning while also improving gut health by modulating gut microbiome [59]. Therefore, the detection of these phytosterols in the studied plant extract underpins its pharmacological value, yet to be fully exploited. The detection of neophytadiene, 1-Heneicosanol, γ-Sitosterol.

Lup-20(29)-en-3-one, and Lupeol implies that the extract possesses anti-inflammatory and antioxidant properties, which, in turn may contribute to its anti-cancer effects [60].

Therefore, considering the ethnomedicinal utilisation of the studied plant in treating PCa and other OS-associated diseases [10], this initial report partly validates its efficacy, and pave the way for further studies that may lead to the isolation and characterisation of lead compounds for the development of safe and potent armamentaria against PCa as well as other OS-related diseases. Considering the preliminary nature of this study, additional investigations using other *in vitro* and *in vivo* models are required to comprehensively appraise this extract's safety and therapeutic potential. Moreover, such investigations will provide crucial empirical data that may guide the translation of research findings into clinical practice.

## 5. Conclusions and recommendations

The ethyl acetate stem bark extract of *B. coriacea* exhibited promising *in vitro* antioxidant and anti-prostate cancer properties. The observed anti-prostate cancer effects are likely mediated through the modulation of key molecular pathways, including the downregulation of androgen receptor (*ar*) and B-cell lymphoma 2 (*bcl-2*), normalization of caspase-3 and cyclin-dependent kinase 1 (*cdk*1), and upregulation of *p53*, ultimately inducing apoptosis. The presence of bioactive phytochemicals in the studied plant extract further underscores its therapeutic potential, and call for more comprehensive investigations using a variety of cancer cell lines and *in vivo* models to facilitate the isolation and empirical characterisation of lead compounds for the development of targeted anticancer therapies. Given its high selectivity toward prostate cancer cells and favorable safety profile, the ethyl acetate stem bark extract of *B. coriacea* is a promising candidate for further pharmacological exploration to unlock its full therapeutic potential.

## Supporting information

**S1 File. Descriptive and inferential statistical data of the DPPH radical scavenging activity ethyl acetate stem bark extract of *B. coriacea*.**
(XLSX)

**S2 File. Descriptive and inferential statistical data of the hydrogen peroxide scavenging activity of ethyl acetate stem bark extract of *B. coriacea*.**
(XLSX)

**S3 File. Descriptive and inferential statistical data of the cytotoxicity of ethyl acetate stem bark extract of *B. coriacea* against Vero CCL-81 cells.**
(XLSX)

**S4 File. Descriptive and inferential statistical data of the inhibitory effects of the ethyl acetate stem bark extract of *B. coriacea* against prostate cancer cells (DU-145).**
(XLSX)

## Acknowledgments

We are grateful for the technical assistance provided by the laboratory technologists, particularly Mr. Daniel Mwaniki (Department of Biochemistry, Microbiology, and Biotechnology, Kenyatta University), Mr. Elias Mandela (Department of Biological Sciences, Mount Kenya University), and Ms. Sally Kamau and Mr. Gilbert Kipkorir (Centre for Traditional Medicine and Drug Research-Kenya Medical Research Institute).

## Author Contributions

**Conceptualization:** Gervason Moriasi, Mathew Ngugi, Peter Mwitari, George Omwenga.

**Data curation:** Gervason Moriasi.

**Formal analysis:** Gervason Moriasi.

**Funding acquisition:** Gervason Moriasi, Mathew Ngugi, George Omwenga.

**Investigation:** Gervason Moriasi.

**Methodology:** Gervason Moriasi, Peter Mwitari.

**Project administration:** Gervason Moriasi.

**Resources:** Gervason Moriasi, Mathew Ngugi, Peter Mwitari, George Omwenga.

**Software:** Gervason Moriasi.

**Supervision:** Mathew Ngugi, Peter Mwitari, George Omwenga.

**Validation:** Gervason Moriasi, Mathew Ngugi, Peter Mwitari, George Omwenga.

**Visualization:** Gervason Moriasi.

**Writing – original draft:** Gervason Moriasi.

**Writing – review & editing:** Gervason Moriasi, Mathew Ngugi, Peter Mwitari, George Omwenga.

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
