## [Decision Letter · Decision Letter 0]

8 Oct 2024

PONE-D-24-33314Antioxidant, anti-prostate cancer potential, and phytochemical composition of the ethyl acetate stem bark extract of Boascia coriacea (Pax.)PLOS ONE

Dear Dr. Moriasi,

Thank you for submitting your manuscript to PLOS ONE. After careful consideration, we feel that it has merit but does not fully meet PLOS ONE’s publication criteria as it currently stands. Therefore, we invite you to submit a revised version of the manuscript that addresses the points raised during the review process.

**The in vitro results show significant antioxidant and anticancer potential, particularly through gene expression modulation (e.g., downregulation of ar and bcl-2, and upregulation of p53), supporting the claims made by the authors. However, the conclusions suggesting the development of novel treatments from this extract are premature without in vivo validation or more comprehensive studies using multiple cancer cell lines. The claims about the presence of bioactive compounds such as lupeol and their contribution to the extract's effects are valid but would benefit from further correlation between their concentrations and the observed biological activities.**

We look forward to receiving your revised manuscript.

Kind regards,

Akingbolabo Daniel Ogunlakin, Phd

Academic Editor

PLOS ONE

**Journal Requirements:**

This study was partly supported by the German Academic Exchange (DAAD) scholarship for PhD Study granted to Gervason Moriasi under the In-Country/In-Region Scholarship Programme (Ref: 91843013).

Reviewers' comments:

Reviewer's Responses to Questions

**Comments to the Author**

1. Is the manuscript technically sound, and do the data support the conclusions?

Reviewer #1: Yes

Reviewer #2: Yes

2. Has the statistical analysis been performed appropriately and rigorously? 

Reviewer #1: Yes

Reviewer #2: Yes

3. Have the authors made all data underlying the findings in their manuscript fully available?

Reviewer #1: Yes

Reviewer #2: Yes

4. Is the manuscript presented in an intelligible fashion and written in standard English?

Reviewer #1: Yes

Reviewer #2: Yes

5. Review Comments to the Author

**Reviewer #1:** Review Comments on the Manuscript "Antioxidant, Anti-Prostate Cancer Potential, and Phytochemical Composition of the Ethyl Acetate Stem Bark Extract of Boascia coriacea (Pax.)"

1. Is the manuscript technically sound, and do the data support the conclusions?

Technical Soundness:

The study employs appropriate methodologies, such as DPPH and hydrogen peroxide scavenging assays for antioxidant potential, MTT assays for cytotoxicity, and qPCR for gene expression analysis, which are standard for such research. However, while the data presented are generally sound, the lack of in vivo experiments limits the direct applicability of the findings to clinical settings. The study relies solely on in vitro assays, which are useful for initial screenings but do not fully support the therapeutic potential of the plant extract as claimed.

Data Support for Conclusions:

The in vitro results show significant antioxidant and anticancer potential, particularly through gene expression modulation (e.g., downregulation of ar and bcl-2, and upregulation of p53), supporting the claims made by the authors. However, the conclusions suggesting the development of novel treatments from this extract are premature without in vivo validation or more comprehensive studies using multiple cancer cell lines. The claims about the presence of bioactive compounds such as lupeol and their contribution to the extract's effects are valid but would benefit from further correlation between their concentrations and the observed biological activities.

2. Has the statistical analysis been performed appropriately and rigorously?

Strength

The statistical methods used (ANOVA, Tukey’s post hoc test, and unpaired t-tests) are appropriate for comparing means across treatment groups and concentrations.

Gene expression data are analyzed using the 2-ΔΔCt method, which is a robust method for relative quantification of qPCR data.

Weaknesses:

There is no mention of tests for normality or homogeneity of variance to validate the assumptions of the parametric tests (ANOVA, t-tests). It is essential to confirm these assumptions were met to ensure the validity of the statistical results.

The manuscript does not report effect sizes or confidence intervals, which would provide more depth and clarity regarding the significance and practical relevance of the findings.

The reporting of p-values is mostly threshold-based (e.g., P < 0.001) rather than providing exact values, which reduces transparency. Providing exact p-values would enhance the rigor of the statistical reporting.

3. Is the manuscript presented in an intelligible fashion and written in standard English?

Strengths:

The manuscript is generally well-written and organized logically, making it relatively easy to follow. The background, methodology, and results are clearly presented, with appropriate figures and tables.

The English is of a high standard, with only a few minor grammatical errors that could be improved for better flow.

Weaknesses:

There are minor language issues in terms of formality. For instance, phrases such as "hence this study" could be rephrased more formally. Some parts of the manuscript are repetitive, particularly in the discussion and conclusion, where certain points are reiterated multiple times without adding new insights.

Recommendations:

Make Raw Data Fully Available: Upload raw qPCR, GC-MS spectra, and statistical data to a public repository, as required by PLOS ONE.

Statistical Rigor: Improve the statistical reporting by testing and reporting the assumptions of parametric tests, providing effect sizes and confidence intervals, and reporting exact p-values.

Clarify Limitations: acknowledge the limitations of using only in vitro methods and emphasize the need for in vivo studies before making strong therapeutic claims.

Language: Revise for grammatical clarity and remove repetitive elements from the discussion and conclusion.

By addressing these points, the manuscript will better meet the standards of PLOS ONE and improve its overall scientific rigor and clarity.

**Reviewer #2:** Review of “Antioxidant, anti-prostate cancer potential, and phytochemical composition of the ethyl acetate stem bark extract of Boascia coriacea (Pax.)”

Oxidative stress is a major factor in the pathogenesis of various diseases including prostate cancer which in particular has presented a significant health challenges, with high morbidity and mortality rates affecting across the globe. Boascia coriacea extract may be used against prostate cancer and other debilitating diseases because of its high free radical scavenging potentials, according to this article. The phytochemical composition of B. coriacea and its effect against prostate cancer through diverse biological pathways were examined in this study. The free radical scavenging potentials of the B. coriacea extract using various antioxidant assays was also examined. This research may help develop natural anti-prostate cancer remedies that improve patient outcomes and clinical application.

The progression from describing the problem to presenting data and discussing potential treatments is logical and well-structured. The below are just few comments and suggestions which need to be addressed;

1. The language and presentation of the manuscript is quite okay, however, there is need to enhance the readability and improve the structure of the manuscript.

2. Proofread so as to eliminate any grammatical errors.

3. Identification and quantification of phytochemical components in B. coriacea would have been preferred using LC-MS or HPLC-MS instead of GC-MS, since the solvent of extraction is moderately polar. Additionally, It has been shown from literature that flavonoids, phenolics, tannins are majorly responsible for the antioxidant potentials of a natural product, and these components can only be detected by LC-MS or HPLC-MS. Kindly consider this in your further works.

4. All other comments have been made on the manuscript. Kindly make the necessary adjustments and corrections.

6. PLOS authors have the option to publish the peer review history of their article (what does this mean?). If published, this will include your full peer review and any attached files.

Reviewer #1: No

Reviewer #2: **Yes: **Thomas Abu

---

## [Author Response · Author response to Decision Letter 0]

12 Oct 2024

Dear Editor and Reviewers,

We thank you for taking your valuable time to review our manuscript. Moreover, we appreciate your positive criticism, strong comments, and suggestions aimed at improving the quality of our research article. We have carefully studied all the review comments and revised our manuscript accordingly. It is our hope that our revised manuscript now meets the approval criteria for publication in this esteemed Journal -PLoS ONE. Please find our responses (highlighted green) to the review comments below.

Editorial comments

We have carefully considered and addressed all the editorial suggestions in the revised manuscript.

Reviewer #1

1. The study employs appropriate methodologies, such as DPPH and hydrogen peroxide scavenging assays for antioxidant potential, MTT assays for cytotoxicity, and qPCR for gene expression analysis, which are standard for such research. However, while the data presented are generally sound, the lack of in vivo experiments limits the direct applicability of the findings to clinical settings. The study relies solely on in vitro assays, which are useful for initial screenings but do not fully support the therapeutic potential of the plant extract as claimed.

Data Support for Conclusions:

The in vitro results show significant antioxidant and anticancer potential, particularly through gene expression modulation (e.g., downregulation of ar and bcl-2, and upregulation of p53), supporting the claims made by the authors. However, the conclusions suggesting the development of novel treatments from this extract are premature without in vivo validation or more comprehensive studies using multiple cancer cell lines. The claims about the presence of bioactive compounds such as lupeol and their contribution to the extract's effects are valid but would benefit from further correlation between their concentrations and the observed biological activities.

Authors’s Response:

Dear Reviewer, thank you for this comment. We concur with your view that additional investigations are required to clarify and substantiate the anticancer effects of the studied plant extract. Consequently, we have clarified this in the revised manuscript and emphasised the need for further studies using other in vitro and in vivo models to provide comphrehensive data, to fully establish the therapuetic potential of this extract. In addition, comphrehensive empirical evidence will foter the traslation of reseaerch findings to into clinical practice. 

We have also corelated the observed antioxidant and antiproliferative effects of the studied plant extract with the concentration of key phytochemicals, especially lupeol, γ-Sitosterol, and Lup-20(29)-en-3-one. We have also added information about the biological activities of these phytochemicals, especially antioxidant, antiinflammatory, and anticancer, and linked them to the observations made in our study. We believe that this prelimary report will pave the way for further studies aimed at demystifying and valorising the the therapeutic potential of the studied plant extract.

2. Has the statistical analysis been performed appropriately and rigorously?

Strength

The statistical methods used (ANOVA, Tukey’s post hoc test, and unpaired t-tests) are appropriate for comparing means across treatment groups and concentrations.

Gene expression data are analyzed using the 2-ΔΔCt method, which is a robust method for relative quantification of qPCR data.

Weaknesses:

There is no mention of tests for normality or homogeneity of variance to validate the assumptions of the parametric tests (ANOVA, t-tests). It is essential to confirm these assumptions were met to ensure the validity of the statistical results.

The manuscript does not report effect sizes or confidence intervals, which would provide more depth and clarity regarding the significance and practical relevance of the findings.

The reporting of p-values is mostly threshold-based (e.g., P < 0.001) rather than providing exact values, which reduces transparency. Providing exact p-values would enhance the rigor of the statistical reporting.

Authors’s Response:

Thank you for this comment, and we apologise for the oversight. We confirm that we performed normality test on the quantitative data from antioxidant and cytotoxicity experiments using the Shapiro-Wilk test, and deemed it normally distributed, and before subjecting it to parametric tests- The revised manuscript also reflects this. 

We have also provided further information on the exact p values and associated information as supplementary files (supporting information, SI 1-4) to ensure transparency as advised. Thank you so much for pointing this out.

3. Is the manuscript presented in an intelligible fashion and written in standard English?

Strengths:

The manuscript is generally well-written and organized logically, making it relatively easy to follow. The background, methodology, and results are clearly presented, with appropriate figures and tables.

The English is of a high standard, with only a few minor grammatical errors that could be improved for better flow.

Weaknesses:

There are minor language issues in terms of formality. For instance, phrases such as "hence this study" could be rephrased more formally. Some parts of the manuscript are repetitive, particularly in the discussion and conclusion, where certain points are reiterated multiple times without adding new insights.

Recommendations:

Make Raw Data Fully Available: Upload raw qPCR, GC-MS spectra, and statistical data to a public repository, as required by PLOS ONE.

Statistical Rigor: Improve the statistical reporting by testing and reporting the assumptions of parametric tests, providing effect sizes and confidence intervals, and reporting exact p-values.

Clarify Limitations: acknowledge the limitations of using only in vitro methods and emphasize the need for in vivo studies before making strong therapeutic claims.

Language: Revise for grammatical clarity and remove repetitive elements from the discussion and conclusion.

By addressing these points, the manuscript will better meet the standards of PLOS ONE and improve its overall scientific rigor and clarity.

Authors’s Response:

To addresss the raised concerns, we have revised the entire manuscript, corrected the grammartical errors, and improved the language quality and fluence. Besides, we have removed the repetitive text from the discussion and conclusion sections and imprived its general quality.

We have provide additional information on p values and associated details as in Excel files supporting information. In addition, we shall deposit all the raw data obtained from this study in an online repository and link it to our manuscript. 

Reviewer #2: 

Review of “Antioxidant, anti-prostate cancer potential, and phytochemical composition of the ethyl acetate stem bark extract of Boascia coriacea (Pax.)”

Oxidative stress is a major factor in the pathogenesis of various diseases including prostate cancer which in particular has presented a significant health challenges, with high morbidity and mortality rates affecting across the globe. Boascia coriacea extract may be used against prostate cancer and other debilitating diseases because of its high free radical scavenging potentials, according to this article. The phytochemical composition of B. coriacea and its effect against prostate cancer through diverse biological pathways were examined in this study. The free radical scavenging potentials of the B. coriacea extract using various antioxidant assays was also examined. This research may help develop natural anti-prostate cancer remedies that improve patient outcomes and clinical application.

The progression from describing the problem to presenting data and discussing potential treatments is logical and well-structured. The below are just few comments and suggestions which need to be addressed;

1. The language and presentation of the manuscript is quite okay, however, there is need to enhance the readability and improve the structure of the manuscript.

2. Proofread so as to eliminate any grammatical errors.

3. Identification and quantification of phytochemical components in B. coriacea would have been preferred using LC-MS or HPLC-MS instead of GC-MS, since the solvent of extraction is moderately polar. Additionally, It has been shown from literature that flavonoids, phenolics, tannins are majorly responsible for the antioxidant potentials of a natural product, and these components can only be detected by LC-MS or HPLC-MS. Kindly consider this in your further works.

Authors’s Response:

Dear reviewer, we appreciate your time and comments aimed at improving the quality of our manuscript. We have carefully considered your suggestions and revised our manuscript. We have corrected the grammatical errors and improved the language of presentation, clarity, readability, and fluency.

Thank you for your suggestion to consider using LC-MS or GC-MS for phytochemical analysis of extracts prepared using polar solvents, in our future studies. We can confirm that this is a reasonable recommendation and will surely take advantage of it. 

Authors’ Remarks

We thank the editor and the reviewers for positive criticism and strong comments to improve our research article’s quality. We invite you to review the revised manuscript and hope that, based on your suggestions, it now meets the standard for approval and publication in this esteemed journal.

We look forward to you receiving your feedback

---

## [Editor Report · Decision Letter 1]

28 Oct 2024

Antioxidant, anti-prostate cancer potential, and phytochemical composition of the ethyl acetate stem bark extract of Boascia coriacea (Pax.)

PONE-D-24-33314R1

Dear Dr. Gervason Moriasi,

We’re pleased to inform you that your manuscript has been judged scientifically suitable for publication and will be formally accepted for publication once it meets all outstanding technical requirements.

Kind regards,

Akingbolabo Daniel Ogunlakin, Phd

Academic Editor

PLOS ONE
---

## [Editor Report · Acceptance letter]

5 Nov 2024

PONE-D-24-33314R1 

PLOS ONE

Dear Dr. Moriasi, 

I'm pleased to inform you that your manuscript has been deemed suitable for publication in PLOS ONE. Congratulations! Your manuscript is now being handed over to our production team.

Kind regards, 

on behalf of

Dr. Akingbolabo Daniel Ogunlakin 

Academic Editor

PLOS ONE